# Multi-Task Neural Network Mapping onto Analog-Digital Heterogeneous Accelerators

**Hadjer Benmeziane[1], Corey Lammie[1], Athanasios Vasilopoulos[1], Irem Boybat[1],**
**Manuel Le Gallo[1], Sidney Tsai[2], Kaoutar El Maghraoui[3], Abu Sebastian[1]**
[1]IBM Research Europe, 8803 Rüschlikon, Switzerland
[2]IBM Research Almaden, 650 Harry Road, San Jose, CA USA
[3]IBM T. J. Watson Research Center, Yorktown Heights, NY 10598, USA
`hadjer.benmeziane@ibm.com`

## Abstract

Multi-Task Learning (MTL) models are increasingly popular for their ability to perform multiple tasks using shared parameters, significantly reducing redundant computations and resource utilization. These models are particularly advantageous for analog-digital heterogeneous systems, where shared parameters can be mapped onto weight-stationary analog cores. This paper introduces a novel framework, entitled Multi-task Heterogeneous Layer Mapping, designed to strategically map MTL models onto an accelerator that integrates analog in-memory computing cores and digital processing units. Our framework incorporates a training process that increases task similarity and account for analog non-idealities using hardware-aware training. In the subsequent mapping phase, deployment on the accelerator is optimized for resource allocation and model performance, leveraging feature similarity and importance. Experiments on the COCO, UCI, and BelgiumTS datasets demonstrate that this approach reduces model parameters by up to 3× while maintaining performance within 0.03% of task-specific models.

## 1 Introduction

Recent advances in the emerging paradigm of In-Memory Computing (IMC) have propelled it as a candidate to overcome the limitations of traditional computing. Analog IMC (AIMC) is of particular interest as it has the potential to scale to higher computational density with improved energy efficiency [1], making it especially appealing for a wide range of applications [2, 3, 4]. By performing computations directly within the memory, AIMC reduces data movement and accelerates computation. However, the inherent noise and variability in analog processing can pose challenges to achieving consistent accuracy [5]. Limited IMC weight capacity and oversized models can prohibit model deployment in a full weight-stationary manner, which is the key to its advantages [6]. As a result, heterogeneous accelerators, which integrate both digital and analog components, offer an effective solution [7], combining the precision and flexibility of digital computation with the energy-efficiency of AIMC. Combining the best of both worlds, heterogeneous accelerators are a strong candidate for future AI systems, both in the edge and in data centers.

With the increasing demand for AI on edge devices, developing efficient methods to optimize and reduce model sizes has become more critical than ever. To address some of the edge-related challenges, MTL [8] has emerged as a powerful approach, enabling a single model to perform multiple tasks simultaneously using the same input representation, thereby minimizing redundant computations and resource consumption. This is especially vital in use cases like autonomous driving, where models must handle tasks such as object detection, semantic segmentation, and decision-making in real time. Such scenarios highlight the growing importance of MTL for delivering high performance while maintaining the efficiency required for edge deployment.

38th Second Workshop on Machine Learning with New Compute Paradigms at NeurIPS 2024(MLNCP 2024).

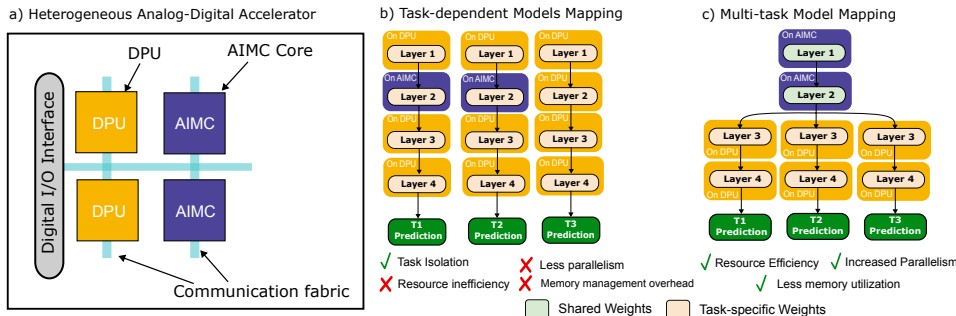

Figure 1: Illustration of (a) a heterogeneous analog-digital accelerator, depicting (b) a traditional heterogeneous approach and (c) a MHLM mapping approach. In the traditional case, heterogeneous mapping is performed independently for each task, whereas for the MHLM case, heterogeneous mapping are performed for a single task-agnostic DNN.

In this paper, we introduce a novel framework, entitled *Multi-task Heterogeneous Layer Mapping (MHLM)*, for training and deployment of MTL models on heterogeneous accelerators with both Digital Processing Units (DPUs) and AIMC components. Our framework focuses on model mapping on heterogeneous analog-digital accelerators. To maximize energy-efficiency, a weight-stationary approach is employed, where all shared components are mapped to AIMC cores, and task-specific components are assigned to DPUs. Shared weights remain stationary on the AIMC cores throughout the computation, significantly reducing the costly data movement between memory and processing units which is required by DPUs. Meanwhile, task-specific components are handled by DPUs to maintain the high accuracy required for specialized tasks, ensuring an optimal balance between performance and energy-efficiency.

We simulate heterogeneous deployment using Phase Change Memory (PCM)-based AIMC cores, modeled with the AIHWKit [9] and DPUs. We demonstrate that MHLM can reduce the number of parameters by $3\times$ while maintaining performance within 0.03% of task-independent models on average, across three different tasks and multiple multi-task models.

## 2   Related Work

**Multi-Task Learning (MTL) Models**   MTL [8, 10] is a sub-field of Machine Learning (ML) where multiple tasks are learned simultaneously using a shared model, leveraging task commonalities to improve learning efficiency, data utilization, and reduce overfitting [11]. Current State-of-the-Art (SOTA) MTL models are often handcrafted, requiring extensive experimentation to determine which components should be shared across tasks, leading to sub-optimal performance. To address these challenges, automated approaches such as Neural Architecture Search (NAS) [12, 13, 14] and adaptive optimizations [14, 15] have been developed, aiming to dynamically discover optimal sharing strategies during training. While these methods can reduce the manual effort involved in designing MTL models and potentially improve scalability, they often add significant computational complexity, increased memory requirements, and poor noise-resiliency in heterogeneous analog-digital accelerators.

**Mapping Strategies for Heterogeneous Analog-Digital Accelerators**   Mapping ML models onto heterogeneous accelerators presents a unique set of challenges, which has spurred significant research efforts in recent years [16, 17]. Traditional approaches often involve splitting the model into layers or modules that can be either efficiently or accurately executed on either analog or digital components [16], optimizing the deployment for a given target in accuracy and efficiency on a workload. However, these strategies have primarily focused on single-task models, with no exploration of how MTL models can be mapped onto such accelerators.

## 3   Multi-task Heterogeneous Layer Mapping

We develop a framework that trains a given network on a set of tasks, optimizing for maximum weight reuse and deployment on analog hardware. It subsequently maps the network onto a heterogeneous accelerator, searching for the largest contiguous part of the network, starting from its first layer, that can be shared between the tasks with minimal loss in accuracy. In detail, our contributions are as follows:

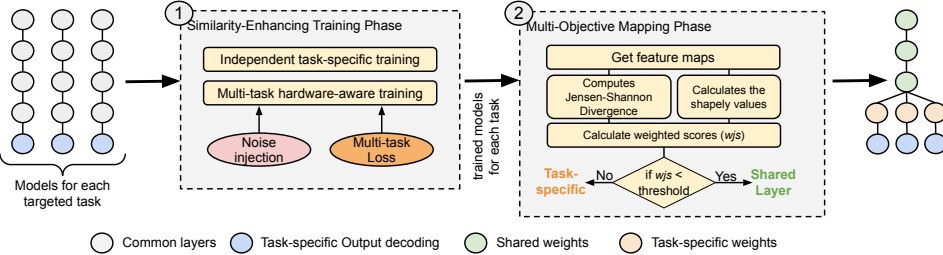

Figure 2: Overview of the MHLM model mapping approach, which includes a ① training phase, where noise injection and joint multi-task loss are used to enhance similarity across tasks, followed by a ② mapping phase, that decides which layer are shared or task-specific.

1. A Hardware-aware (HWA) training algorithm that enhances the similarity of model weights across tasks, enabling more efficient resource sharing and reducing redundancy;
2. An adaptive post-training mapping algorithm that uses Jensen-Shannon Divergence (JSD) and Shapley values to dynamically search for the largest shared part of the network, while keeping model performance over a threshold, maximizing the shared parameters and thus the energy efficiency;
3. A comprehensive evaluation on benchmark datasets, including COCO, UCI, and BelgiumTS, demonstrating the effectiveness of our approach in improving resource utilization, reducing energy consumption, and enhancing overall performance in multi-task learning scenarios.

The two components of our framework, namely the training and mapping algorithms are presented in Fig. 2. We start with a ① training phase, where Gaussian noise is injected into the model weights to simulate the variability found in analog computing environments. During this phase, a joint multi-task loss function encourages the similarity of features across tasks. The next ② mapping phase uses JSD and Shapley values [18] to evaluate the similarity and importance of features, determining whether a feature should be shared across tasks, i.e., mapped on AIMC cores, or remain task-specific, i.e., mapped on DPUs.

## 3.1 Similarity-Enhancing Training

① involves a HWA training algorithm that injects Gaussian noise into the model's weights during training. This noise simulates the variability encountered in analog computing, encouraging the model to learn more robust and similar representations across tasks. However, the added noise increases feature dissimilarity, rendering standard multi-task learning methods ineffective in the context of analog deployment. To address this, we propose a novel training approach that includes a joint multi-task loss to explicitly enhance similarity among tasks by penalizing large differences between the feature distributions of different tasks. The training algorithm pseudo-code is provided in Alg. 1.

After injecting noise during the forward propagation passes and obtaining the outputs for all tasks, the algorithm calculates a floating-point joint multi-task loss $\mathcal{L}_{\text{total}}$ (L9). This loss comprises two components: the task-specific loss $\mathcal{L}_{\text{task}}$ for each task and a regularization term that penalizes large differences between the feature distributions $\mathbf{z}_i$ and $\mathbf{z}_j$ of different tasks using the KL divergence. The regularization term is weighted by a factor, $\lambda$. Once the total loss is computed, gradients are accumulated (L10), and the model weights are updated (L11) to minimize the loss[1].

The benefits of this training approach are illustrated in Fig. 3, where the performance different training strategies are compared using the COCO dataset. Since analog devices are prone to temporal variations, causing their performance to fluctuate over time [19], we report the 1-day performance after the devices are programmed for all experiments[2]. This is reported after each training epoch for

---

[1]KL divergence was selected for training due to its computational efficiency and simplicity, while Jensen-Shannon Divergence (JSD) is employed for mapping because of its symmetric properties, offering a more balanced and robust measure of similarity between task-specific feature distributions.

[2]The 1-day accuracy metric is chosen to balance the need to assess early-stage drift impacts. Accuracy typically decreases linearly with respect to logarithmic time.

**Algorithm 1** Training for Multi-task Model Mapping
___
**Require:** Number of tasks $T$, Gaussian noise level $\sigma$, joint multi-task loss weight $\lambda$
**Require:** Number of epochs $T_{\text{epoch}}$
 1: **for** each epoch $t = 1, \ldots, T_{\text{epoch}}$ **do**
 2:     Get input data $\mathbf{x}$ and task labels $\mathbf{y}_t$ for all tasks $t \in \{1, \ldots, T\}$
 3:     Clear gradients, `optimizer.zero_grad()`
 4:     **for** each task $t \in \{1, \ldots, T\}$ **do**
 5:         Apply Gaussian noise to weights: $\mathbf{W}_t = \mathbf{W}_t + \mathcal{N}(0, \sigma^2)$
 6:         Get task output and encoded features $\hat{\mathbf{y}}_t, \mathbf{z}_t = f_t(\mathbf{x}; \mathbf{W}_t)$
 7:     **end for**
 8:     Compute joint multi-task loss:

$$\mathcal{L}_{\text{total}} = \sum_{t=1}^{T} \mathcal{L}_{\text{task}}(\hat{\mathbf{y}}_t, \mathbf{y}_t) + \lambda \sum_{i<j} \text{KL}(\mathbf{z}_i || \mathbf{z}_j)$$

 9:     Accumulate gradients, $\mathcal{L}_{\text{total}}$.backward()
10:     Update model weights, `optimizer.step()`
11: **end for**
___

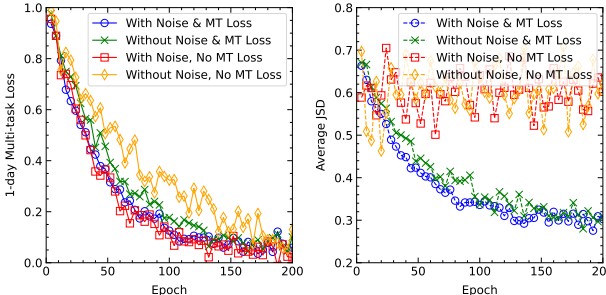

Figure 3: Evaluation of Training Strategies on the COCO Dataset. The plot shows how the average loss decreases across epochs using multi-task training with a lambda value of 0.05.

four different training strategies: with and without noise injection, and with and without the joint multi-task loss. When the joint multi-task loss is applied, the JSD decreases, reflecting improved similarity between the learned representations of the tasks. Without the joint loss, the JSD remains around 0.6, indicating that the tasks are less aligned in their feature distributions. The inclusion of Gaussian noise aids in reducing variability, contributing to more stable and similar representations across tasks.

### 3.2 Multi-Objective Mapping Algorithm

For ②, the objective is twofold: (i) maximize the shared portion of the model, enabling its deployment in weight-stationary AIMC, and (ii) simultaneously maximize the average 1-day performance across all tasks. The mapping algorithm uses JSD to measure the similarity between the feature map distributions of different tasks and Shapley values to assess the importance of each feature map.

The adaptive mapping algorithm is described in Supplementary Alg.2. The core decision-making process balances these metrics, governed by a threshold $\tau$, which determines whether a feature map should be shared or remain task-specific. Additionally, a weighting factor $\beta$ is introduced to prioritize configurations that enhance the 1-day performance. These threshold are empirically set. Supplementary Fig.2 shows the impact of these threshold on the final average performance.

## 4 Experiments

### 4.1 Experimental Setup

**Datasets:** We evaluate the performance of our method using three datasets – COCO [20], UCI [21] and an autonomous driving scenario with BelgiumTS [22].

Table 1: 1-day Performance Results on COCO. SS: Semantic Segmentation, OD: Object Detection, IC: Image Classification.

| Model | Training Scenario | 1-day mIoU (SS) | 1-day mAP (OD) | 1-day Accuracy (IC) | Shared Portion (%) | Analog Params (M) | Digital Params (M) | Analog MAC Ops (%) [▽] |
|---|---|---|---|---|---|---|---|---|
| **MTL-NAS** | – | 0.692 ± 0.050 | 0.635 ± 0.040 | 0.822 ± 0.045 | 22 | 10.2 | 34.9 | 30% |
| **EDNAS** | – | 0.688 ± 0.045 | 0.628 ± 0.045 | 0.820 ± 0.040 | 31 | 11.4 | 24.3 | 32% |
| **ResNet50 [25]** | Task-specific w/o HWA* | 0.753 | 0.686 | 0.858 | 0 | 0 | 76.1 | 0% |
| | Task-specific w/ HWA | 0.743 ± 0.050 | 0.673 ± 0.040 | 0.852 ± 0.050 | 0 | 76.1 | 0 | 98% |
| | AdaShare | 0.712 ± 0.048 | 0.640 ± 0.042 | 0.810 ± 0.048 | 58 | 17.9 | 10.8 | 48% |
| | AdaMTL | 0.690 ± 0.045 | 0.610 ± 0.043 | 0.792 ± 0.045 | 75 | 18.3 | 11.0 | 55% |
| | **MHLM** | **0.739** ± 0.040 | **0.668** ± 0.040 | **0.846** ± 0.050 | **65** | **16.25** | **9.9** | **65%** |
| **DETR [26]** | Task-specific w/o HWA* | 0.762 | 0.702 | 0.878 | 0 | 0 | 122.6 | 0% |
| | Task-specific w/ HWA | 0.751 ± 0.050 | 0.691 ± 0.050 | 0.873 ± 0.040 | 0 | 122.6 | 0 | 86% |
| | AdaShare | 0.720 ± 0.041 | 0.655 ± 0.042 | 0.835 ± 0.041 | 66 | 28.7 | 12.2 | 52% |
| | AdaMTL | 0.695 ± 0.044 | 0.620 ± 0.044 | 0.803 ± 0.043 | 80 | 29.4 | 12.5 | 56% |
| | **MHLM** | **0.748** ± 0.040 | **0.688** ± 0.040 | **0.867** ± 0.040 | **72.5** | **28.8** | **10.6** | **68%** |
| **FocalNet [27]** | Task-specific w/o HWA* | 0.738 | 0.678 | 0.848 | 0 | 0 | 85.4 | 0% |
| | Task-specific w/ HWA | 0.732 ± 0.060 | 0.671 ± 0.060 | 0.841 ± 0.050 | 0 | 85.4 | 0 | 92% |
| | AdaShare | 0.710 ± 0.058 | 0.636 ± 0.057 | 0.805 ± 0.052 | 48 | 15.6 | 13.5 | 44% |
| | AdaMTL | 0.685 ± 0.059 | 0.610 ± 0.059 | 0.795 ± 0.058 | 70 | 15.8 | 13.8 | 58% |
| | **MHLM** | **0.725** ± 0.060 | **0.666** ± 0.050 | **0.838** ± 0.050 | **53** | **14.6** | **13.0** | **76%** |

* Full digital models are not susceptible to conductance drift or noise.
▽ Batch Norm and non weight-stationary attention MACs are included in the computation of this percentage.

**Comparison Methods** We compare our results to NAS methods including MTL-NAS [23] and ED-NAS [24], and adaptive sharing methods such as AdaShare [14] and AdaMTL [15]. For each of these methods, we apply a HWA on the final multi-task network. We use the same mapping, i.e., shared portion in analog. We also compare to the original full task-specific networks with and without HWA. Full digital baselines for the adaptive multi-task networks can be found in Supplementary Table 1. Supplementary Section F expands on the training hyperparameter for each model.

**Evaluation Metrics:** For a comprehensive assessment, we employ a range of evaluation metrics across the different tasks. For the object detection and segmentation tasks, we use mean Average Precision (mAP) and mean Intersection over Union (mIoU) as primary metrics. For classification tasks, accuracy is used to measure the effectiveness of our approach. Additionally, we report the shared portion of the model, which quantifies the proportion of the network (in the number of parameters) that is shared across tasks, providing insights into the trade-offs between resource efficiency and task-specific performance.

**Experiment Mapping Time:** The training process with hardware-aware training and joint multi-task loss takes about $1.4\times$ longer than conventional training, due to similarity enhancement, but remains manageable as it is a one-time process. The mapping process, which calculates JSD for shared portions, averages around 15 minutes for larger networks such as DETR and FocalNet.

## 4.2 Results

**COCO:** The results highlight the substantial reduction in the number of parameters achieved by our MHLM framework compared to task-specific training. Across all models, MHLM uses up to 3x fewer parameters while maintaining performance within 1% of task-specific training. The increase in the shared portion in MHLM directly correlates to energy savings, as more of the model is deployed on analog components, which are more energy-efficient. This trade-off between shared portion and performance is critical for resource-constrained environments. Although AdaMTL offers a higher shared portion, it suffers from a drastic drop in performance due to its failure to account for analog noise, emphasizing the importance of our hardware-aware training. AdaShare performs slightly better but still under-performs compared to MHLM, demonstrating the effectiveness of our noise-aware approach for hybrid analog-digital platforms. The low standard deviations across the metrics indicate that the performance was consistently high across multiple runs.

**UCI:** The results presented in Fig. 4 provide a detailed comparison of accuracy and shared portion for different tasks for the UCI dataset under different conditions. Fig. 4(a) highlights the accuracy across tasks for the baseline (0% shared) trained with and without noise, as well as for the multi-task mapping method (MHLM) under similar noise conditions. The baseline without noise shows the highest accuracy, but it does not benefit from sharing, which limits resource efficiency. Introducing noise in the baseline configuration results in a noticeable drop in accuracy across most tasks, indicating the sensitivity of the models to analog noise.

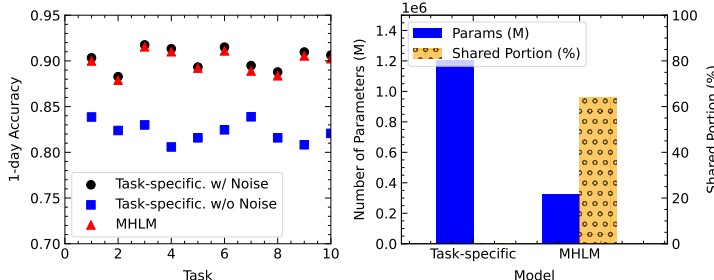

Figure 4: Task-wise comparison of (a) accuracy and (b) the shared portion for the UCI dataset.

Table 2: 1-day Performance Results on BelgiumTS Dataset. SS: Sign Segmentation, SC: Sign Classification.

| Model | Training Scenario | 1-day mIoU (SS) | 1-day Accuracy (SC) | Shared Portion (%) | Analog Params (M) | Digital Params (M) | Analog MAC Ops (%) [▽] |
|---|---|---|---|---|---|---|---|
| **ViT-Adapter-S [28]** | Task-specific w/o HWA | 0.810 | 0.948 | 0 | 0 | 36.9 | 0% |
| | Task-specific w/ HWA* | 0.803 ± 0.007 | 0.940 ± 0.007 | 0 | 36.9 | 0 | 95% |
| | AdaShare | 0.773 ± 0.007 | 0.910 ± 0.008 | 58 | 12.3 | 10.1 | 50% |
| | AdaMTL | 0.760 ± 0.006 | 0.900 ± 0.007 | 72 | 12.7 | 10.2 | 52% |
| | **MHLM** | **0.800** ± 0.006 | **0.938** ± 0.007 | **60** | **12.3** | **10.1** | **65%** |
| **MaskFormer [29]** | Task-specific w/o HWA | 0.820 | 0.955 | 0 | 0 | 43.5 | 0% |
| | Task-specific w/ HWA* | 0.813 ± 0.005 | 0.948 ± 0.006 | 0 | 43.5 | 0 | 88% |
| | AdaShare | 0.780 ± 0.006 | 0.910 ± 0.007 | 62 | 14.5 | 11.1 | 54% |
| | AdaMTL | 0.768 ± 0.006 | 0.900 ± 0.007 | 70 | 15.0 | 11.5 | 57% |
| | **MHLM** | **0.810** ± 0.005 | **0.945** ± 0.006 | **62** | **14.5** | **11.1** | **70%** |
| **MHA-JAM [30]** | Task-specific w/o HWA | 0.800 | 0.940 | 0 | 0 | 35.1 | 0% |
| | Task-specific w/ HWA* | 0.793 ± 0.007 | 0.930 ± 0.008 | 0 | 35.1 | 0 | 90% |
| | AdaShare | 0.760 ± 0.007 | 0.900 ± 0.008 | 61 | 11.7 | 8.5 | 48% |
| | AdaMTL | 0.748 ± 0.007 | 0.890 ± 0.008 | 75 | 11.9 | 8.7 | 50% |
| | **MHLM** | **0.790** ± 0.007 | **0.928** ± 0.007 | **61** | **11.7** | **8.5** | **68%** |

\* Full digital models are not susceptible to conductance drift or noise.

[▽] Batch Norm and non weight-stationary attention MACs are included in the computation of this percentage.

Conversely, the MHLM approach without noise demonstrates improved accuracy compared to the baseline with noise, showcasing the effectiveness of our method in sharing components while still delivering strong performance. When noise is introduced to MHLM, a slight decrease in accuracy is observed, but it remains competitive with the baseline without noise, underscoring the robustness of the method. Fig. 4(b) shows the shared portion across tasks, where the MHLM configurations achieve significant sharing without substantial drops in performance. The results emphasize the advantage of our approach in balancing resource efficiency with task performance, making it well-suited for deployment in noise-prone environments.

**BelgiumTS:** Maximizing shared portions resulted in better resource utilization, for real-time sign segmentation and detection, particularly in edge-like models. The performance drops were minimal, with low standard deviations. The 1-day performance metrics underscore the robustness of these models under time-constrained conditions, with only slight reductions in accuracy compared to full training (0.005).

## 5 Discussion & Conclusion

The overall results demonstrate the efficacy of our MHLM framework. Notably, models trained with MHLM consistently exhibited a high shared portion while maintaining robust performance metrics. The Pareto analysis, shown in Supplementary Fig.1, further emphasizes the strategic trade-offs between performance and resource sharing, revealing how our framework adeptly balances these objectives. The ablation study, Supplementary Table 3, highlights the critical role of each step in MHLM.

MHLM is particularly impactful in scenarios with stringent resource constraints, enabling significant shared component usage without substantial performance degradation. Note that our framework can be generalized for optimized deployment in mixed-precision solely digital accelerators, but it is out of the scope of this work. Future work will quantitatively determine the energy-efficiency using system-level simulations and explore the joint architecture search and mapping for heterogeneous analog-digital accelerators.

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

# A   Multi-Objective Analysis Results

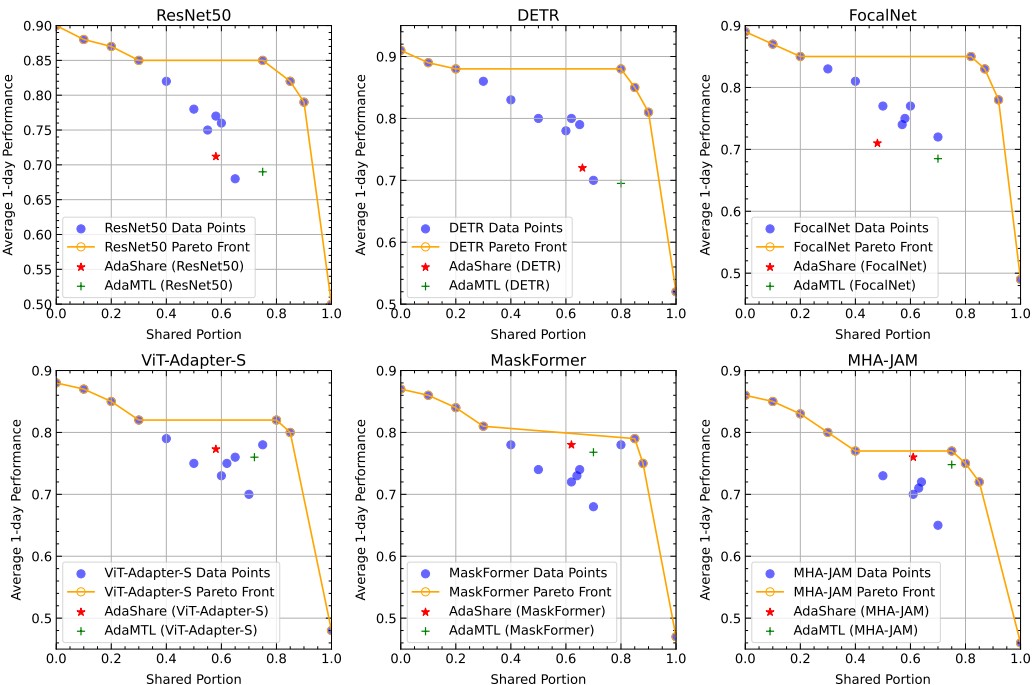

Supplementary Figure 1: Pareto fronts for different models across COCO and BelgiumTS datasets. Each subplot shows the trade-off between the shared portion and average performance for a specific model, with the Pareto front (orange line) indicating the optimal configurations. Blue dots represent data points near the Pareto front, providing additional context.

The Pareto fronts for the six models evaluated across the COCO and BelgiumTS datasets are shown in Supplementary Fig. 1. Each subplot corresponds to a specific model and illustrates the trade-off between the shared portion and average performance. The orange line in each subplot represents the Pareto front, which highlights the optimal configurations where both the shared portion and average performance are maximized. Additionally, the figure includes blue crosses representing non-Pareto points that are close to the Pareto front. These points provide further insight into the trade-offs, showing configurations that are suboptimal compared to the Pareto-optimal configurations. By visualizing how small changes in the shared portion can impact average performance, the figure underscores the importance of balancing the shared portion with task-specific performance when designing multi-task models for deployment in hybrid analog-digital systems.

The Pareto front is extracted by running the mapping algorithm, Supplementary Alg.2, under different configurations to balance resource sharing and 1-day performance. For each solution, the algorithm computes a weighted JS divergence that reflects task similarity and feature importance. Based on a threshold, the algorithm determines whether a layer is shared or task-specific, with all subsequent layers marked as task-specific once the threshold is exceeded.

# B Multi-Objective Mapping Algorithm

Supplementary Alg.2 shows MHLM mapping pseudo-code. The weighted JSD (Weighted_JS) is designed to balance the trade-off between maximizing shared components across tasks while maintaining high task-specific performance. The equation incorporates two factors: the Shapley value for each feature map and the 1-day average performance. The Shapley value, influenced by the scaling factor $\alpha$, reflects the importance of the feature map across multiple tasks, ensuring that critical features with high task contribution are less likely to be shared. This mechanism prevents performance degradation by discouraging the sharing of highly specialized features. The inclusion of the 1-day performance term, modulated by the weighting factor $\beta$, ensures that the mapping process accounts for analog drift and noise over time. By emphasizing performance stability after 1 day of operation, the algorithm prioritizes robustness in hardware-deployed models. The parameters $\alpha$ and $\beta$ were empirically selected through cross-validation experiments.

---

**Supplementary Algorithm 2** Multi-Objective Mapping Algorithm

---

**Require:** Feature maps for each task $F_t$ where $t \in \{1, 2, \ldots, T\}$
**Require:** Scaling factor $\alpha$ for Shapley value influence
**Require:** Threshold $\tau$ for sharing decision
**Require:** Weighting factor $\beta$ for 1-day performance maximization
**Ensure:** Sets of shared parts and task-specific parts: Shared_Parts, Task_Specific_Parts
 1: Initialize Shared_Parts $\leftarrow \emptyset$, Task_Specific_Parts $\leftarrow \emptyset$
 2: **for** each feature map $k$ **do**
 3:      Initialize Avg_JS$^k$ as the average JS divergence across task pairs
 4:      Compute the Shapley value Shapley_Value$^k$ for feature map $k$
 5:      Compute the weighted divergence Weighted_JS$^k$ = Avg_JS$^k \times (1 + \alpha \times$ Shapley_Value$^k) \times$ $(1 + \beta \times$ 1-day Average Performance )
 6:      **if** Weighted_JS$^k < \tau$ **then**
 7:          Add $k$ to Shared_Parts
 8:      **else**
 9:          Add $k$ and all subsequent layers to Task_Specific_Parts for all tasks
10:          **break** loop to stop further shared selection
11:      **end if**
12: **end for**
13: **return** Shared_Parts, Task_Specific_Parts

---

## C Full Digital Baselines

Supplementary Table 1 and 2 show the results of training the multi-task learning models fully on digital, i.e., without any noise. These are the maximum performances possibly acheived by each of the model.

Supplementary Table 1: Full digital baselines for AdaShare, AdaMTL, and MHLM on COCO Dataset.

| Model | Method | mIoU (SS) | mAP (OD) | Accuracy (IC) |
|---|---|---|---|---|
| | AdaShare | 0.782 | 0.673 | 0.854 |
| ResNet50 | AdaMTL | 0.791 | 0.688 | 0.855 |
| | MHLM | 0.741 | 0.671 | 0.849 |
| | AdaShare | 0.781 | 0.694 | 0.884 |
| DETR | AdaMTL | 0.810 | 0.679 | 0.873 |
| | MHLM | 0.750 | 0.689 | 0.869 |
| | AdaShare | 0.724 | 0.657 | 0.829 |
| FocalNet | AdaMTL | 0.710 | 0.643 | 0.818 |
| | MHLM | 0.729 | 0.667 | 0.840 |

Supplementary Table 2: Full digital baselines for AdaShare, AdaMTL, and MHLM on BelgiumTS Dataset.

| Model | Scenario | mIoU (SS) | Accuracy (SC) |
|---|---|---|---|
| ViT-Adapter-S | AdaShare | 0.792 | 0.929 |
| | AdaMTL | 0.785 | 0.920 |
| | MHLM | 0.802 | 0.939 |
| MaskFormer | AdaShare | 0.798 | 0.930 |
| | AdaMTL | 0.792 | 0.920 |
| | MHLM | 0.811 | 0.947 |
| MHA-JAM | AdaShare | 0.780 | 0.920 |
| | AdaMTL | 0.768 | 0.910 |
| | MHLM | 0.798 | 0.928 |

## D Ablation Study

Supplementary Table 3: Ablation study results.

| Model | Training Scenario | Avg. 1-day Performance | Shared Portion (%) |
|---|---|---|---|
| | Task-specific Training | 0.85 ± 0.05 | 0 |
| | Without Noise Injection | 0.68 ± 0.06 | 55 |
| ResNet50 | Without MT Loss | 0.71 ± 0.05 | 30 |
| | Without Feature Importance | 0.83 ± 0.04 | 35 |
| | Full MHLM | 0.84 ± 0.04 | 65 |
| | Task-specific Training | 0.88 ± 0.04 | 0 |
| | Without Noise Injection | 0.70 ± 0.06 | 58 |
| DETR | Without MT Loss | 0.74 ± 0.04 | 35 |
| | Without Feature Importance | 0.86 ± 0.04 | 40 |
| | Full MHLM | 0.87 ± 0.04 | 72.5 |
| | Task-specific Training | 0.87 ± 0.06 | 0 |
| | Without Noise Injection | 0.69 ± 0.06 | 57 |
| FocalNet | Without MT Loss | 0.72 ± 0.05 | 33 |
| | Without Feature Importance | 0.85 ± 0.04 | 37 |
| | Full MHLM | 0.86 ± 0.04 | 70 |

The ablation study results, summarized in Supplementary Table 3, reveal the impact of key components on model performance and resource sharing. Independent training yields the highest task-specific performance but with no shared resources. When noise injection is omitted, performance drops significantly, highlighting its importance for robustness. Without the MTL, the model achieves slightly better results than without noise, yet still with a low shared portion, indicating the critical role of MT loss in enabling component sharing. Removing feature importance maintains high performance but reduces the shared portion, underscoring its contribution to efficient sharing. The full approach, integrating all components, strikes the best balance between performance and resource sharing, demonstrating the effectiveness of our methodology.

## E    Impact of JSD threshold on mapping

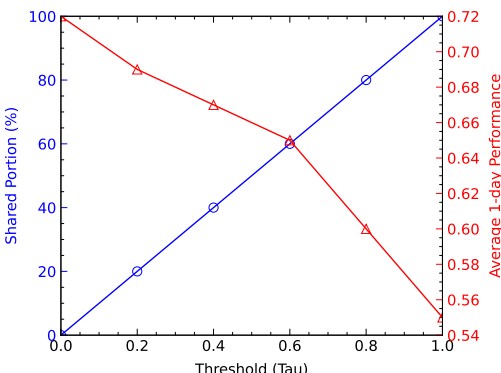

Supplementary Figure 2: Impact of JSD Threshold on Shared Portion and 1-Day Performance.

The impact of varying the JSD divergence threshold on the proportion of shared components and the average 1-day performance is illustrated in Supplementary Fig. 2. The two objectives in the multi-objective mapping algorithm—maximizing the shared portion of the model while maintaining high 1-day performance—are inherently contradictory. As the threshold increases, more parts of the model are shared, leading to significant reductions in model parameters, thus improving resource efficiency. However, this comes at the expense of performance. As observed in the figure, with lower values, the 1-day performance remains close to the maximum value of 0.72, but the shared portion is minimal. Conversely, as it approaches 1, the shared portion of the model increases towards 100%, but the 1-day performance drops below 0.60.

## F    Training Hyperparameters

Each network in our experiments was trained with specific hyperparameters, obtained with hyperparameter optimization. Table 4 summarizes the key hyperparameters used during training for each model. For COCO datasets, a resizing to 224x224 and random cropping were applied.

Supplementary Table 4: Training hyperparameters for each network. Learning rate (LR), Batch size (BS), Scheduler, and Number of epochs (Epochs) are listed.

| Model | LR | BS | Scheduler | Epochs |
|-------|-----|-----|-----------|--------|
| ResNet50 (COCO) | 0.001 | 16 | Cosine Annealing | 100 |
| DETR (COCO) | 0.0005 | 8 | StepLR with step size 30 | 100 |
| FocalNet (COCO) | 0.0003 | 16 | Cosine Annealing | 100 |
| ViT-Adapter-S (BelgiumTS) | 0.0005 | 32 | Cosine Annealing with Warm Restarts | 200 |
| MaskFormer (BelgiumTS) | 0.0003 | 32 | StepLR with step size 50 | 200 |
| MHA-JAM (BelgiumTS) | 0.001 | 32 | Cosine Annealing | 200 |

The following AIHWKit 'rpu_config' was used for all experiments where analog in-memory cores were deployed. This configuration simulates hardware non-idealities, such as noise and variability

in the Phase Change Memory (PCM)-based cores, while maintaining performance close to that of fully digital implementations.

```python
def create_rpu_config(g_max=25, tile_size=512, modifier_std=0.07):
    rpu_config = InferenceRPUConfig()

    rpu_config.mapping.digital_bias = True
    rpu_config.mapping.weight_scaling_omega = 1.0
    rpu_config.mapping.weight_scaling_columnwise = True
    rpu_config.mapping.learn_out_scaling = True
    rpu_config.mapping.out_scaling_columnwise = True
    rpu_config.mapping.max_input_size = tile_size
    rpu_config.mapping.max_output_size = tile_size

    rpu_config.noise_model = PCMLikeNoiseModel(g_max=g_max)
    rpu_config.remap.type = WeightRemapType.CHANNELWISE_SYMMETRIC
    rpu_config.clip.type = WeightClipType.FIXED_VALUE
    rpu_config.clip.fixed_value = 1.0

    rpu_config.modifier.type = WeightModifierType.MULT_NORMAL
    rpu_config.modifier.rel_to_actual_wmax = True
    rpu_config.modifier.std_dev = modifier_std
    rpu_config.forward = IOParameters()
    rpu_config.forward.out_noise = 0.05
    rpu_config.forward.inp_res = 1 / (2**8 - 2)
    rpu_config.forward.out_res = 1 / (2**8 - 2)
    rpu_config.drift_compensation = GlobalDriftCompensation()
    return rpu_config
```

