# OpenReview forum: "Multi-Task Neural Network Mapping onto Analog-Digital Heterogeneous Accelerators"
_NeurIPS.cc/2024/Workshop/MLNCP — MLNCP Poster_

### Official Review · Reviewer_Qa1s · 2024-10-01
**Optimal parameter sharing across tasks for models on analog-digital heterogenous systems**

**Rating:** 7
**Confidence:** 3

**Review:**

This paper attempts multiple things at once: The idea is to identify as large a model as can be shared across tasks to reduce the overall parameter count of a multi-task (multi-modal?) model. The model is then trained using a mixture of task-specific and multi-task losses. Importantly, the model is trained using hardware-aware training to prepare for acceleration on an analog compute device. After training, the largest connected part of the network is identified that can be shared across tasks. The shared-part of the model is then moved to the analog domain to maximally use the available acceleration. The authors report a drop in total required parameters of ~3x.

The paper is straight forward to understand, the problem setting and proposed solution make sense. The results, especially the plateau in the pareto-fronts showing a near-constant performance over a large share of shared parameters is interesting.

The paper would be stronger if the results were translated into overall gains in compute efficiency. A concern with heterogeneous digital-analog systems is that due to Amdahl's law, the overall gain through analog compute can be small (2x? 3x?) - at least when compared to the high cost of new hardware which usually require gains north of 10x.

The idea to maximize the analog portion of the model and minimize A-D conversions is a good direction of thinking, numbers on the overall achievable gain would contribute to the impact of the study.

---

### Official Review · Reviewer_L34g · 2024-10-03
**A framework for multi-task learning using analog accelerators, supported by simulations**

**Rating:** 6
**Confidence:** 2

**Review:**

This paper is about Multi-Task Learning (MTL), that is learning multiple tasks simultaneously using a shared model. It combines MTL with analog in-memory computing (AIMC), where shared parameters are mapped onto AIMC cores, and task-specific components are assigned to digital processors.

The paper includes simulations of a model based on Phase Change Memory (PCM)-based AIMC cores, modeled with AIHWKit. Experiments are performed on the COCO, UCI, and BelgiumTS datasets. The paper demonstrates that the proposed framework uses up to 3x fewer parameters while maintaining performance within 1% of task-specific models.



**Advantages of the approach**

Using a shared model is useful for edge devices where the physical size of the chip matters. This is especially useful in situations where inferences for the different tasks are performed synchronously: model sharing reduces the amount of computation and thus of energy consumption. Mapping the shared parameters onto AIMC components makes the approach potentially even more energy-efficient.



**Questions**

The paper argues that one advantage of using a shared model for multiple tasks is to reduce overfitting (line 53), but I wonder if this is really a genuine advantage of the MTL approach. The fact that the proposed method performs slightly less well than the baseline (training task-specific models) shows that it actually likely *underfits*. This of course would apply to any MTL method, not just the MHLM method proposed in this work.

Once the model architecture is chosen (step 2), do you need to retrain the final model architecture? If not, it’s not clear to me how the weights of the final architecture are chosen.

Figure 4, left. Do you have an explanation why the performance of “MHLM” and “Task-specific with noise” seem to coincide so well?

Do you have any estimates of energy savings?

---

### Decision · Program_Chairs · 2024-10-10

Accept (Poster)